

# Unlocking the potential of pollarded oaks: A 375–year hydroclimate reconstruction from northcentral Spain

Alba Sanmiguel-Vallelado[1], Max C. A. Torbenson[2], Jan Esper[2,3], Gabriel Sangüesa-Barreda[1], Carlos Prado-López[4], José Miguel Olano[1]

[1]EiFAB-iuFOR, Universidad de Valladolid, Soria, 42004, Spain
[2]Department of Geography, Johannes Gutenberg University, Mainz, 55099, Germany
[3]Global Change Research Institute, Czech Academy of Sciences, Brno 603 00, Czech Republic
[4]Fundación para la Investigación del Clima, Madrid, 28040, Spain

*Correspondence to*: Alba Sanmiguel-Vallelado (alba.sanmiguel@uva.es)

**Abstract.** Pollarded trees—traditionally pruned and maintained for centuries near rural settlements—represent an untapped resource for climate reconstruction in Mediterranean lowlands. In this study, we evaluate the potential of 102 pollarded deciduous oaks from two communal woodlands (*dehesas*) in northcentral Spain as proxies for past hydroclimatic variability. Using the correlation between latewood and November–June precipitation, we reconstruct regional precipitation variability from 1649 to 2023, achieving calibration/verification correlations of 0.71–0.83 against regional and large-scale instrumental datasets. The reconstruction reveals pronounced interannual to multidecadal variability, with precipitation ranging from 250 mm to 815 mm. The longest dry spell lasted 25 years (1818–1842), while the wettest sustained period extended over 29 years (1953–1981). We identify 14 extremely dry years (< 298 mm) and 24 extremely wet years (> 592 mm). Extreme droughts during the pre-instrumental period coincide with historical records, such as Catholic *pro pluvia* rogations—ceremonies traditionally held in response to agricultural drought—in 1683, 1698, 1734, 1737, 1738, 1775, 1868 and 1898. Our findings demonstrate that pollarded trees, when sampled from sites with asynchronous management, preserve robust climate signals and provide reliable high-resolution information on precipitation variability across Mediterranean *dehesas*.

## 1 Introduction

Precipitation variability, at both subseasonal to decadal timescales, is a primary driver of ecosystem productivity in Mediterranean environments (Bartsch et al., 2020; Gaona et al., 2022; Madrigal-González et al., 2017). Droughts, in particular, have long been a major concern, as evidenced by early documentary sources such as Arabic, Christian, and Byzantine chronicles and records (Brázdil et al., 2018; Dominguez-Castro and García-Herrera, 2016; Meklach et al., 2021). In fact, an increasing number of studies have linked extreme drought events to abrupt societal changes across the Mediterranean, from Ancient Greece to Medieval Spain (Camuera et al., 2023; Christian and Elbourne, 2018; Kaniewski et al., 2013). Although Mediterranean societies may have become more resilient to drought impacts (Savelli et al., 2022), reconstructing past precipitation variability remains essential for placing current changes (IPCC, 2023) into a broader temporal perspective and



for distinguishing natural variability from anthropogenic forcing (Esper et al., 2024). This is particularly relevant considering the high uncertainty that still surrounds future precipitation projections (Hawkins and Sutton, 2011; Robinson et al., 2021; Rowell, 2012), and the fact that the Mediterranean region is recognized as a climate change hotspot (Giorgi, 2006).

However, long-term reconstructions of hydroclimate variability in lowland regions are still underrepresented, particularly in Mediterranean areas such as the Iberian Peninsula. To date, most dendrochronological reconstructions rely on mountain forests, for instance in the Central System (Ruiz-Labourdette et al., 2014) and the Iberian System (Esper et al., 2015; Tejedor et al., 2016, 2017), where tree growth mainly reflects late spring and summer precipitation. Yet, the high spatial variability of rainfall in mountainous areas limits the extrapolation of these data to lower elevations (Ljungqvist et al., 2020). Although several

Mediterranean tree species, such as *Pinus pinaster* Ait. and *Pinus halepensis* Mill., show strong growth sensitivity to precipitation (Bogino and Bravo, 2008; Tejedor et al., 2020), a major limitation is the scarcity of long-lived individuals due to drought and human pressure (Peñuelas and Sardans, 2021; Piovesan and Biondi, 2021).

Pollarded oaks growing in *dehesas* could help fill this gap, as they are long-lived, highly sensitive to precipitation, and widespread in low- to mid-elevation areas close to human settlements (Olano et al., 2023). These culturally and ecologically

emblematic open woodlands have been shaped over centuries by traditional practices such as pollarding and grazing (Butler, 2013; Harrison, 1996). Despite meeting key criteria of climate proxies, *dehesas* have been largely overlooked due to their regular management. Paradoxically, the same management has helped preserve ancient trees up to the present day, typically growing under low inter-tree competition, which enables the development of multi-centennial tree-ring chronologies (Camarero and Valeriano, 2023; Olano et al., 2023; Rozas, 2005). Pollarded trees exhibit robust precipitation signals spanning

from prior autumn to current spring (Olano et al., 2023), a climatically critical period that encompasses key ecological and socio-economic phases, particularly those linked to staple crop production in nearby areas (Vicente-Serrano et al., 2013). Altogether, *dehesas* represent a valuable and yet underused resource for reconstructing long-term hydroclimatic variability in a region where high-quality and long climate records remain scarce (Leal et al., 2015).

Pollarding implies removing the upper part of the tree through systematic pruning at varying intensities (Petit and Watkins,

2003), traditionally aimed at obtaining essential provisioning materials such as poles for construction, charcoal, fodder for livestock and firewood (Ayanz, 1994; Moreno and López-Díaz, 2009). The management often leads to the formation of internal cavities with increasing tree age (Remm and Lõhmus, 2011), which poses a challenge for sampling and limits the development of long chronologies. Pollarding produces abrupt reductions in tree-ring width due to the interruption of latewood production immediately after cutting (Schweingruber, 2007). A previous study in deciduous oak *dehesas* of northcentral Spain found that

this effect lasts on average about three years at a median pollarding interval of 21 years (Sanmiguel-Vallelado et al., 2024). The resulting growth anomalies have long been considered an obstacle for dendroclimatic research, as they were assumed to weaken the climatic signal. However, recent studies suggest that climatic signals can be preserved if pollarding occurred asynchronously among trees (Olano et al., 2023). As such, the pollarding effect may be seen as stochastic noise in the traditional dendroclimatic perspective of tree growth (Cook and Kairiukstis, 2013).



The main objective of this study was to use deciduous pollarded oaks for a formal climate reconstruction. To this end, we developed and analyzed tree-ring width chronologies based on total ring width (RW), as well as earlywood and latewood widths (EW and LW, respectively) of *Quercus faginea* Lam. and *Q. pyrenaica* Willd. growing in two nearby communal woodlands of northcentral Spain. We pursued four objectives: (1) determine the species' growth sensitivity to precipitation; (2) assess to what extent traditional pollarding management alters the precipitation signal, both at the individual tree level and

stand level; (3) evaluate the robustness and geographical reach of a low-elevation precipitation reconstruction model from pollarded oaks; (4) compare findings with existing large-scale reconstructions and historical documentary records. By presenting the first tree ring-based hydroclimate reconstruction from pollarded oaks, we aim to advance understanding of the sensitivity of these culturally and ecologically important ecosystems.

## 2 Materials and methods

### 2.1 Study area

The study area is located in the Soria province, northcentral Spain (Fig. 1). We sampled two communal woodlands that have been historically managed through asynchronous pollarding (Olano et al., 2023; Sanmiguel-Vallelado et al., 2024): Vilviestre (Vi; 41.88°N, 2.65°W, 1069 m a.s.l., 270 ha) and Valonsadero (Va; 41.81°N, 2.53°W, 1040 m a.s.l., 2793 ha), which are approximately 13 km apart. Both sites lie on sandstone-derived soils and experience a continental climate with Atlantic

influences. Winters are typically cold with frequent frosts, while summers are hot, and the region is characterized by strong diurnal and seasonal temperature contrasts. The studied woodlands are dominated by deciduous oak species, although tree species composition varies between Vi where *Quercus faginea* Lam. is more abundant, and Va where *Q. pyrenaica* Willd. dominates. Data from both woodlands were combined to construct a single chronology for each ring component: total ring width, earlywood width, and latewood width.




**Figure 1: (a) Distribution of *Q. pyrenaica* and *Q. faginea* in Europe and Northern Africa (Caudullo et al., 2017). (b) Location of the Vilviestre (Vi) and Valonsadero (Va) *dehesas* northwest of Soria, showing the central grid cell points from the Climate Research Unit (CRU) and *Fundación para la Investigación del Clima* (FIC) used to obtain climate data. Digital Terrain Model source: PNOA 1:200,000 ETRS89 HU30 Soria. (c) Pollarded *Q. faginea* trees in Vi. (d) Walter & Lieth climate diagram for Vi constructed from climate data from the FIC. The red line represents mean monthly temperature, and the blue line represents mean monthly precipitation. Periods where precipitation falls below twice the temperature are classified as arid (shaded with red dots), while periods above this threshold are considered wet (shaded with blue lines).**



## 2.2 Tree-ring data

Fieldwork was conducted between October 2022 and June 2024. Given the low competition and strong common signal among trees in the *dehesas*, we extracted only one core per tree using a Pressler increment borer (Olano et al., 2023). To reduce the potential influence of synchronous management events, trees were sampled across the whole extent of each woodland. In total, 102 trees were sampled (62 in Vi, 40 in Va), comprising 45 *Q. pyrenaica* and 57 *Q. faginea* individuals. Tree cores were air-dried, mounted on wooden supports, and sanded with progressively finer grades ranging from 80 to 800 grit. High-resolution images were obtained using a CaptuRING device (García-Hidalgo et al., 2022), equipped with a Tokina Macro 100 mm f/2.8 lens mounted on a Nikon D7500 camera. Images were stitched using the PTGui software (New House Internet Services BV, Netherlands) resulting in a final resolution of 5,897 dpi. Total ring width (RW), earlywood width (EW), and latewood width (LW) were measured manually using the CooRecorder software (Cybis Elektronik & Data AB, Sweden) at a precision of 0.004 mm. Cross-dating accuracy was assessed using the xDateR Shiny app.

To isolate the climate signal in tree-ring data, age-related growth trends were removed from RW, EW, and LW using a cubic smoothing splines with a 50% frequency cutoff set to two-thirds of the series length (Cook, 1985; Fritts, 1976). This procedure was implemented using the dplR package in R (Bunn, 2008, 2010; R Core Team, 2024). Ring width index (RWI), earlywood width index (EWI), and latewood width index (LWI) were calculated as the ratios between the observed values and fitted curves. All detrended series were then aggregated into residual chronologies by removing the remaining serial autocorrelation from a standard chronology previously computed using the bi-weight robust mean method (Cook, 1985). Chronology reliability was evaluated using the expressed population signal (EPS), calculated over 30-year windows lagged by 15 years. Chronologies were truncated to periods where EPS > 0.85, a commonly accepted threshold indicating sufficient signal strength for reliable population-level inference (Wigley et al., 1984).

## 2.3 Climate data

Monthly precipitation data for the study area were obtained from two sources: the Climate Research Unit (CRU) TS4.08 dataset (1901–2022; 0.5° resolution; Harris et al., 2020) and the *Fundación para la Investigación del Clima* (FIC, www.ficlima.org) database. The FIC database provides high-resolution gridded data (1 km²; 1951–2020) based on observed series from 5,577 precipitation and 2,515 temperature observatories across the Iberian Peninsula and the Balearic Islands. These series are interpolated monthly using a multivariable thin plate spline method (Duchon, 1976) that accounts for altitude, distance from the sea, longitude, latitude, and mountain range orientation. We used both datasets for subsequent analyses: FIC offers a high-resolution interpolated model, while CRU provides longer temporal coverage and is widely used in climate studies (Gebrechorkos et al., 2025; Kaser et al., 2010; Torbenson et al., 2025). Precipitation values from both sources were extracted for the grid point closest to Vi (Fig. 1).





**2.5 Statistical analysis**

**2.5.1 Detection of climate signals in tree-ring chronologies**

Correlation analyses between the RWI, EWI, and LWI chronologies and monthly/seasonal precipitation were performed using the treeclim package in R (Zang and Biondi, 2015). In consideration of lagged climate effects on tree growth (Fritts, 1976), both current-year and previous-year data were considered. We assessed the consistency of the climate signal over time, i.e., its

temporal stability, by calculating moving correlations using 20-year windows. Additionally, spatial patterns are assessed by correlating the tree-ring chronologies with gridded CRU precipitation data across the Mediterranean basin.

**2.5.2 Evaluation of the influence of pollarding on climate-growth relationships**

Previous evidence indicates that growth reductions induced by pollarding diminish the climate signal in tree-ring series (Olano et al., 2023). We applied the method proposed by Sanmiguel-Vallelado et al. (2024) to identify pollarding events in individual

tree-ring index series. This method uses a Random Forest model specifically trained to detect growth suppression and recovery patterns characteristic of pollarding.

We assessed the effect of pollarding events on the precipitation signal recorded in tree growth using CRU precipitation data. The procedure includes fitting three separate linear mixed-effects models (LMMs), one for each growth index (RWI, EWI, and LWI). The response variable in each model was the standardized tree-level growth index series. Fixed effects included the

interaction between (1) time since the pollarding event—categorized as year 0 (the event itself), years 1 to 9, and year 10 (which includes all subsequent years)—and (2) the precipitation during the period most strongly correlated with each chronology (Sect. 2.5.1). Random effects accounted for repeated measures within individual trees.

To assess whether the pollarding-related signal loss observed at the tree level scales up to the stand-scale signal, we removed from each growth index series all post-pollarding years in which the climatic signal was significantly weakened. The filtered

series were then used to build new chronologies following the same procedures described in Sect. 2.2. We subsequently recalculated and compared the strength of the precipitation signal between these chronologies and the original ones.

**2.5.3 Calibration and validation of the reconstruction model using independent instrumental data**

We reconstructed past precipitation for the period 1649–2023 from tree-ring data using a linear regression model to define the transfer function, with the LWI chronology as the predictor and prior November–current June precipitation from the closest

grid point as the response variable. This season was identified as the strongest among all tested combinations of significant months (Fig. S4). We adopted a split calibration–validation approached, combining FIC data for late calibration (1962–2020) and CRU data for early validation (1902–1961), to test the robustness of the reconstruction across independent datasets. Model performance was evaluated using standard verification statistics: Pearson's correlation coefficient (r), coefficient of determination ($R^2$), reduction of error (RE), coefficient of efficiency (CE), and root-mean-square error (RMSE). Positive RE

and CE values indicate model skill (Fritts, 1976). The final reconstruction model was calibrated over the full FIC period (1952–



2020) to maximize temporal coverage and data quality. Prediction intervals at the 95% confidence level were calculated using the *predict()* function from R's base stats package. To correct distributional biases and better capture extreme values in the reconstruction, we applied the Quantile Mapping (QM) bias correction method (Gudmundsson et al., 2012; Robeson et al., 2020) using the RQUANT algorithm implemented in the qmap R package (Gudmundsson, 2016). Additionally, the final model was compared against the full CRU precipitation record (1902–2022) to assess consistency.

### 2.5.4 Benchmarking against large-scale hydroclimate reconstructions and historical documentary records

To evaluate the reliability and regional consistency of our reconstruction, we compared it with several independent large-scale hydroclimatic reconstructions (Table 2). When applicable, series were extracted for the grid point closest to our study area: specifically, the 41.75°N, 2.75°E grid point for the European precipitation reconstruction (Pauling et al., 2006), the 40.74°N, 2.5°E grid point for the Paleo Hydrodynamics Data Assimilation product (PHYDA, Steiger et al., 2018), and the 42.25°N, 2.75°E grid point for the Great Eurasian Drought Atlas (GEDA, Cook et al., 2024). For reconstructions based on precipitation (rather than indices), seasonal values were summed to approximate the November–June seasonal window used in this study.

The reconstructed November–June precipitation covers the critical period for crop development in the region, a key socioeconomic sector strongly dependent on water availability throughout the growing season (Vicente-Serrano et al., 2020). Therefore, extremely dry years in the reconstruction likely reflect agricultural drought conditions, which we defined as years with reconstructed precipitation anomalies exceeding –1.5 standard deviations (Tejedor et al., 2017). Conversely, extremely wet years were defined as those exceeding +1.5 standard deviations, while multi-year dry or wet periods were defined as sequences of ten or more consecutive years with precipitation consistently below or above the mean (Niu et al., 2024).

To evaluate the reliability of these extreme values, we used historical records of *pro pluvia* rogations—Catholic ceremonies held to request rain during droughts affecting agriculture or livestock. These events are considered a robust proxy for pre-instrumental agricultural droughts in Spain (Dominguez-Castro and García-Herrera, 2016). From the rogations compiled by Domínguez-Castro et al. (2021), we selected 131 documented events corresponding to 81 different years between the period 1649 and 1929. These ceremonies took place in winter and spring across several localities surrounding the study area (latitude: 41º–43º; longitude: –6.5º to –3.0º), consistent with the regional representativeness of our chronology (Fig. 2). The number of events held in each location is indicated in parentheses: Alaejos (1), Cabreros del Monte (1), Cuéllar (2), La Seca (2), Medina del Campo (1), Tordesillas (14), Valladolid (42), and Zamora (68). We then qualitatively checked whether any of the *pro pluvia* rogations occurred during reconstructed extremely dry years and whether these were corroborated by the Catalogue of Historical Droughts in Spain (Centro de Estudios Hidrográficos, 2013). For the instrumental period (1901–2023), we also considered drought episodes recorded in the Spanish Drought Catalogue (Trullenque-Blanco et al., 2024). Finally, we performed a quantitative test using a Monte Carlo bootstrap analysis to assess whether reconstructed precipitation anomalies in rogation years were significantly lower than expected by chance. This involved resampling 81 random years from the 1649–1929 period over 10,000 iterations and fitting a normal distribution to the resulting anomalies to calculate two-tailed confidence intervals.





## 3 Results

### 3.1 Developed chronologies

A summary of the main descriptive statistics of the raw ring-width series (RW, EW and LW) and the detrended series (RWI, EWI and LWI) is presented in Table 1. All these metrics indicate that the sampled trees share a very strong common climate signal, higher and more temporally coherent for total ring-width and latewood components. Accordingly, the final truncated chronologies (EPS > 0.85) span 375 years (1649–2023) for RWI and LWI with 102 sample size (Fig. S1 and S3), while the EWI chronology extends over a shorter period of 255 years (1769–2023) (Fig. S2). Running EPS values for detrended series are reported in Appendix A (Tables S1–S3).

**Table 1: Descriptive tree-ring statistics. AR1 is first order autocorrelation. The mean correlation between trees (RBAR), signal-to-noise ratio (SNR), and expressed population signal (EPS) are calculated for detrended data.**

| Wood component | Mean series length (years) | Mean width ± SD (mm) | Mean AR1 | RBAR | SNR | EPS |
|---|---|---|---|---|---|---|
| Total ring-width | | 0.94 ± 0.59 | 0.59 | 0.25 | 33.75 | 0.97 |
| Earlywood | 276 | 0.55 ± 0.27 | 0.50 | 0.12 | 13.88 | 0.93 |
| Latewood | | 0.39 ± 0.35 | 0.47 | 0.24 | 31.27 | 0.97 |

### 3.2 Precipitation signal of the chronologies

Tree growth was primarily limited by moisture availability during and prior to the growing season, showing a strong positive correlation with winter–spring precipitation (Fig. S4). The LWI chronology exhibited the strongest association with November–June precipitation (r = 0.83, p < 0.001; Fig. S4). Precipitation from November to June accounted for an average of 75.64% of the total annual precipitation, considering the hydrological year.

FIC and CRU precipitation series were highly correlated (r = 0.89, p < 0.001) and produced nearly identical growth–climate relationships during their overlapping period (RWI: r = 0.81–0.82; EWI: r = 0.56; LWI: r = 0.83; Fig. S5). These minor differences in correlations likely reflect the different spatial resolutions of the FIC and CRU datasets. However, when considering the full temporal extent of each dataset, the FIC series more effectively captured the climate signal embedded in the chronologies (r = 0.83; Fig. S3).

The spatial field correlation analysis confirmed that the LWI chronology captures a regional precipitation signal (1951–2022; Fig. 2a). The strongest correlations (r > 0.6) occurred across central and western Spain, excluding the Mediterranean seaboard. Significant but weaker correlations (r = 0.4–0.5) are also evident in western and southern France. Correlations performed for the earlier period (1902–1950; Fig. 2b) are spatially noisier and slightly weaker, reflecting the lower reliability of early instrumental data, but they remain significant and regionally consistent.



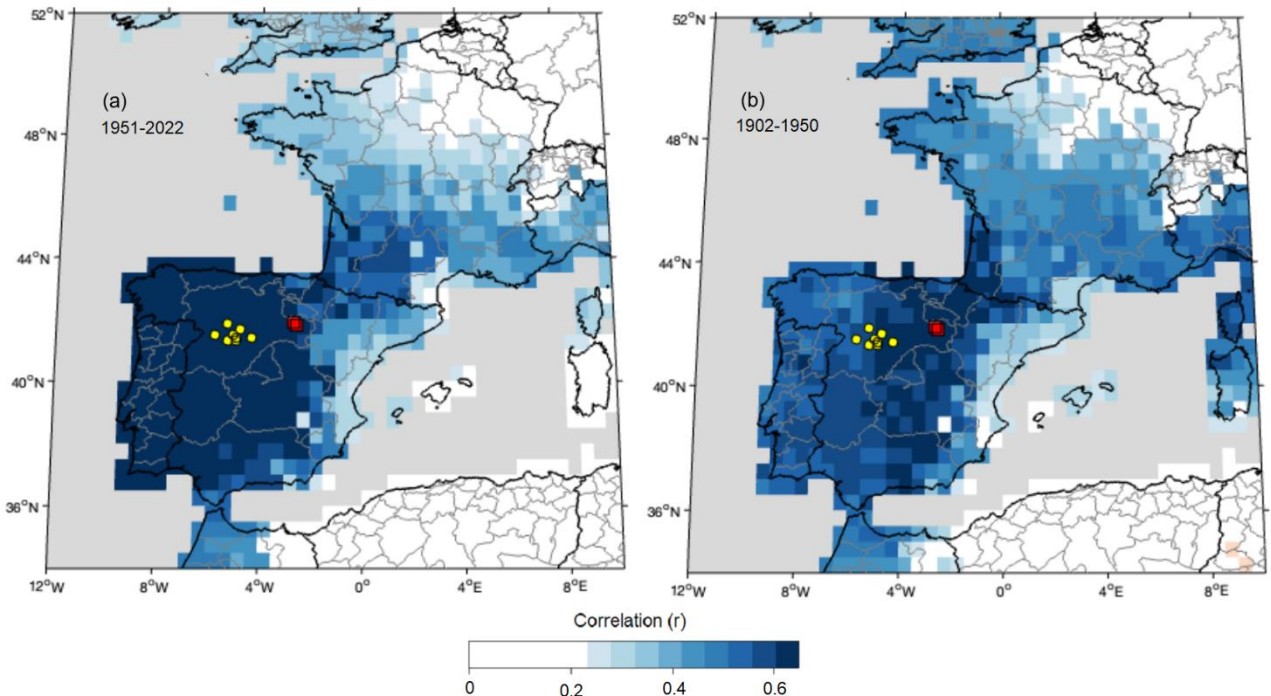

**Figure 2: Spatial field correlations (p < 0.05) between November–June precipitation (CRU) and the LWI chronology for two periods: (a) 1951–2022 and (b) 1902–1950. Yellow dots indicate the locations of the towns where *pro pluvia* rogation ceremonies were held, used here to validate extremely dry years. Red squares indicate the locations of the Vi and Va sampled *dehesas*.**

### 3.3 Influence of pollarding on climate-growth relationships

Pollarding influenced the climate signal at the tree level. The climate sensitivity of the RWI series significantly declined in the second year after pollarding, while in the LWI series this effect persisted for three years (Fig. 3a; Table S4–S6). During this period, growth in both the RWI and LWI series ceased to respond to climatic variations (Fig. 3b). However, from the eighth year onward, the climate sensitivity in both series surpassed pre-pollarding levels (Fig. 3a; Table S4–S6). This increased sensitivity caused the growth of pollarded trees to reflect climatic variations in an amplified manner, showing stronger responses to drought episodes (reduced growth) as well as to periods of abundant rainfall (enhanced growth) compared to pre-pollarding conditions (Fig. 3b). In contrast, the EWI series did not show significant changes in its climate signal following pollarding (Fig. 3a; Table S5).





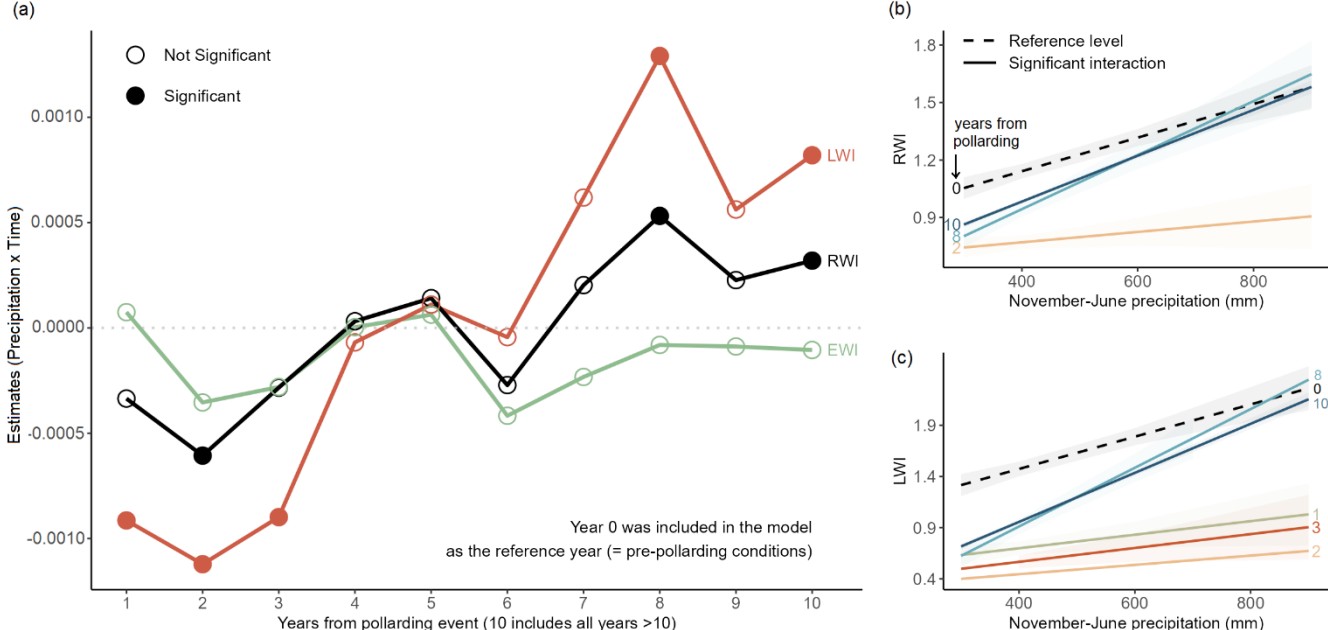

**Figure 3: (a) Changes in the seasonal precipitation signal following pollarding events (November–June for RWI and LWI; November–February for EWI). Years with signals significantly different from pre-pollarding conditions (p < 0.05) are marked with**

235 **filled circles. (b–c) Significant interaction effects between November–June precipitation and time since pollarding on tree growth indices: (b) RWI and (c) LWI. Each line represents a different year since the pollarding event, with dashed lines indicating the reference year (Year 0) and solid lines representing years with significant interactions (p < 0.05). Shaded areas represent 95% confidence intervals. Precipitation data source: CRU (1902–2022).**

Despite this temporary reduction in climate signal at the tree level, filtering the affected years prior to building the stand-level chronologies (Fig. S6)—1 year in RWI and 3 years in LWI—did not enhance their climate sensitivity (Fig. 4) or improve their temporal stability (Fig. S7–S8). Correlations between the RWI chronology and CRU precipitation data (1902–2022) remained stable before and after filtering (r = 0.74), as did correlations with FIC precipitation data (1952–2020; r = 0.82) (Fig. 4). The LWI chronology showed similar patterns, with correlations against CRU data consistently at 0.76 pre- and post-filtering, and

correlations against FIC data increasing slightly from 0.83 to 0.84 after filtering (Fig. 4). These results indicate that pollarding did not compromise the climatic signal at the stand level, supporting the robustness of these chronologies for climate reconstruction without correction.




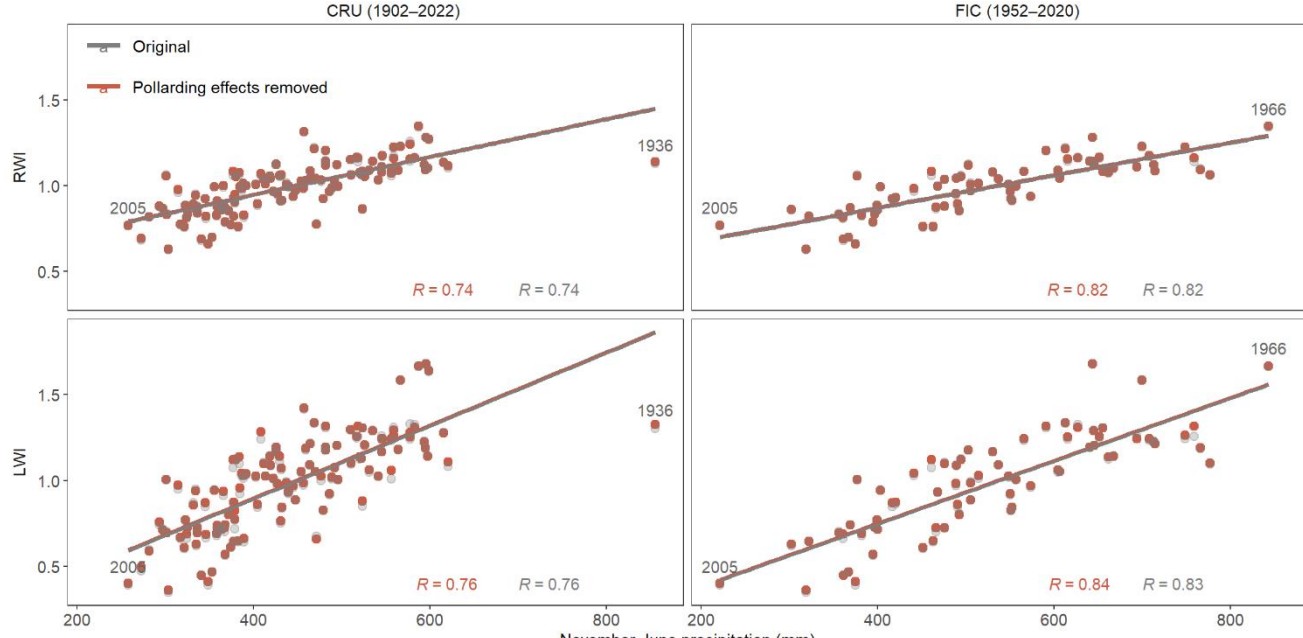

**Figure 4: Scatterplots detailing the November–June precipitation signal in RWI (top) and LWI (bottom) regional chronologies, before (grey) and after (red) removing the effects of pollarding. Left panels: CRU data (1902–2022); right panels: FIC data (1952–2020).**

### 3.3 Precipitation reconstruction and validation with independent instrumental data

Calibration against FIC precipitation data for 1962–2020 yielded an $R^2$ of 0.69, while validation against CRU data for 1902–1961 produced a correlation of r = 0.71 (p < 0.001; Fig. 5a–c). Positive values for both reduction error (RE = 0.45) and coefficient of efficiency (CE = 0.45) further support the model's reliability. The RMSE of 76 mm indicates an acceptable prediction error relative to observed variability (Fig. 5a). Moreover, the reconstructed and observed series displayed similar variability (SD = 97 vs. 104 mm) and frequency distributions (Fig. 5d–5e).





**Figure 5: (a) Split calibration (FIC 1962–2020) and validation (CRU 1902–1961) results of the November–June precipitation reconstruction derived from pollarded oak LWI data in northcentral Spain. The red curve shows observed precipitation totals; grey dotted and solid curves show the uncorrected and bias-corrected reconstructions (QM), respectively. The shaded area represents the 95% prediction interval of the calibration model. (b-c) Scatterplots of linear regressions between the LWI chronology and FIC precipitation (1962–2020; calibration) and between CRU precipitation and the bias-corrected reconstruction (1902–1961; validation). (d–e) Frequency distributions of bias-corrected reconstructed and CRU precipitation, respectively (1902–1961).**

To maximize both temporal coverage and data quality, the final model was calibrated over the full 1952–2020 period using FIC data ($R^2$ = 0.70; Fig. S9a–S9b) and applied to reconstruct November–June precipitation from 1649 to 2023. The reconstructed series was further compared against the full CRU dataset (1902–2022), showing a strong correlation (r = 0.76; Fig. S9c) and closely matching frequency distributions (Fig. S9d–S9e), reinforcing the robustness of the reconstruction.



**3.4 Validation of the reconstruction using previous large-scale reconstructions**

Several independent hydroclimate reconstructions were used to assess the consistency of our reconstructed November–June precipitation series spanning 1649−2023 (Table 2). Over the full overlapping periods, all reconstructions showed statistically significant correlations with our series. The strongest agreements were found with the GEDA (r = 0.41), following by regional indices such as the Western Mediterranean precipitation reconstruction (r = 0.37), the Ebro Valley drought index (r = –0.36) and the Andalusia precipitation index (r = 0.31). By contrast, the weakest correlations were obtained with large-scale products,

including PHYDA (r = 0.13 and 0.14 for MAM and JJA, respectively) and the European gridded precipitation reconstruction (r = 0.25). When analysed by century, all reconstructions—except PHYDA—showed significant correlations with our series during the 18th and 19th centuries. The highest agreements were observed with the Ebro Valley drought index (r = −0.50) in the 18th century and with the Andalusia precipitation index (r = 0.32) in the 19th century. Results for the 17th and 20th centuries were more variable, with the strongest correlations found with the Ebro Valley drought index (r = −0.50) and the

GEDA (r = 0.54), respectively.

**Table 2: Hydroclimate records used for correlation with the new November–June precipitation reconstruction since 1649 and four centuries. Pearson correlations are reported unless otherwise noted (see footnotes 2–3). Statistically significant values (p < 0.05) are indicated with an asterisk (\*). IO refers to instrumental observations, DS to documentary sources, NP to natural proxies, and PS**
**PHYDA paleoclimate series (reanalysis from multiple proxies and climate-model assimilation; Steiger et al. (2018)). SPEI is the Standardized Precipitation Evapotranspiration Index, DJF the December–February season, MAM the March–May season, and JJA is the June–August season. NA indicates missing century-scale correlations due to a shorter common period between series.**

| Reference | Geographic coverage | Predictors | Target | Season | Correlation (common period) | 17th century | 18th century | 19th century | 20th century |
|---|---|---|---|---|---|---|---|---|---|
| Tejedor et al. (2019)[1,2] | Ebro Valley | DS | Drought index | Annual | −0.36* (1649–1899) | −0.50* | −0.50* | −0.26* | NA |
| Rodrigo et al. (1999)[1,3] | Andalusia | DS | Precipitation index | DJF+MAM | 0.31* (1649–1997) | 0.38* | 0.35* | 0.32* | 0.29* |
| Camuffo et al. (2010)[1] | Western Mediterranean | IO | Precipitation | DJF+MAM | 0.37* (1649–1800) | 0.27 | 0.40* | NA | NA |
| Pauling et al. (2006) | Europe | IO, DS, NP | Precipitation | DJF+MAM | 0.25* (1649–1900) | 0.36* | 0.21* | 0.25* | NA |
| Cook et al. (2024) | Eurasia | NP | PDSI | JJA | 0.41* (1649–2018) | 0.37* | 0.45* | 0.30* | 0.54* |
| Steiger et al. (2018) | Global | PS | SPEI | MAM | 0.13* (1649–2000) | 0.14 | 0.19 | 0.02 | 0.19 |
| Steiger et al. (2018) | Global | PS | SPEI | JJA | 0.14* (1649–2000) | 0.25 | 0.23* | 0.03 | 0.19 |

[1] Accessed via DOCU-CLIM, a global documentary climate dataset for climate reconstructions (Burgdorf et al., 2023).
[2] Categorical index from 0 (no drought) to 3 (severe drought); correlations based on Spearman's rho.



³ Categorical index from −2 (low precipitation) to 2 (high precipitation); correlations based on Spearman's rho.

### 3.5 Validation of the reconstruction extremes using historical documentary records

The reconstructed November–June precipitation for 1649–2023 averaged 445 mm, ranging from 250 mm (1963) to 815 mm (1964) (Fig. 6). A total of 14 extremely dry years (< 298 mm) and 24 extremely wet years (> 592 mm) were identified (Table

S7). Clusters of extremely dry years occurred in 1734–1738 and 1981–2012, whereas extremely wet years were concentrated between 1731 and 1800 (Fig. 6). The longest drought lasted 25 years (1818–1842), and the longest wet period 29 years (1953–1981) (Table S8). The most severe drought, based on mean November–June precipitation, occurred in the 17th century (1652–1669; 407 mm), while the wettest period was in the second half of the 20th century (1953–1970; of 484 mm) (Table S8).

The identification of extremely dry years in the reconstructed series—likely triggers of agricultural droughts—is strongly

supported by documentary evidence. During the instrumental period, extremely dry years coincided with drought episodes reported in the Spanish Drought Catalogue (Trullenque-Blanco et al., 2024). For the pre-instrumental period, they coincided with historical *pro pluvia* rogation records (Domínguez-Castro et al., 2021) and/or were corroborated by the Catalogue of Historical Droughts in Spain (Centro de Estudios Hidrográficos, 2013) (Fig. 7a; Table S7). Quantitative validation of the reconstructed extremely dry years showed that the mean estimated precipitation during the selected 81 rogation years (379

mm) fell outside the 99% confidence interval from Monte Carlo simulations (415–468 mm), indicating that precipitation anomalies in these years were unlikely to occur by chance and were significantly associated with drought events (Fig. 7b). A few of the 81 rogation years overlapped with wetter-than-average conditions early in the series; all of these were spring ceremonies, primarily recorded in Zamora (Fig. 7a).



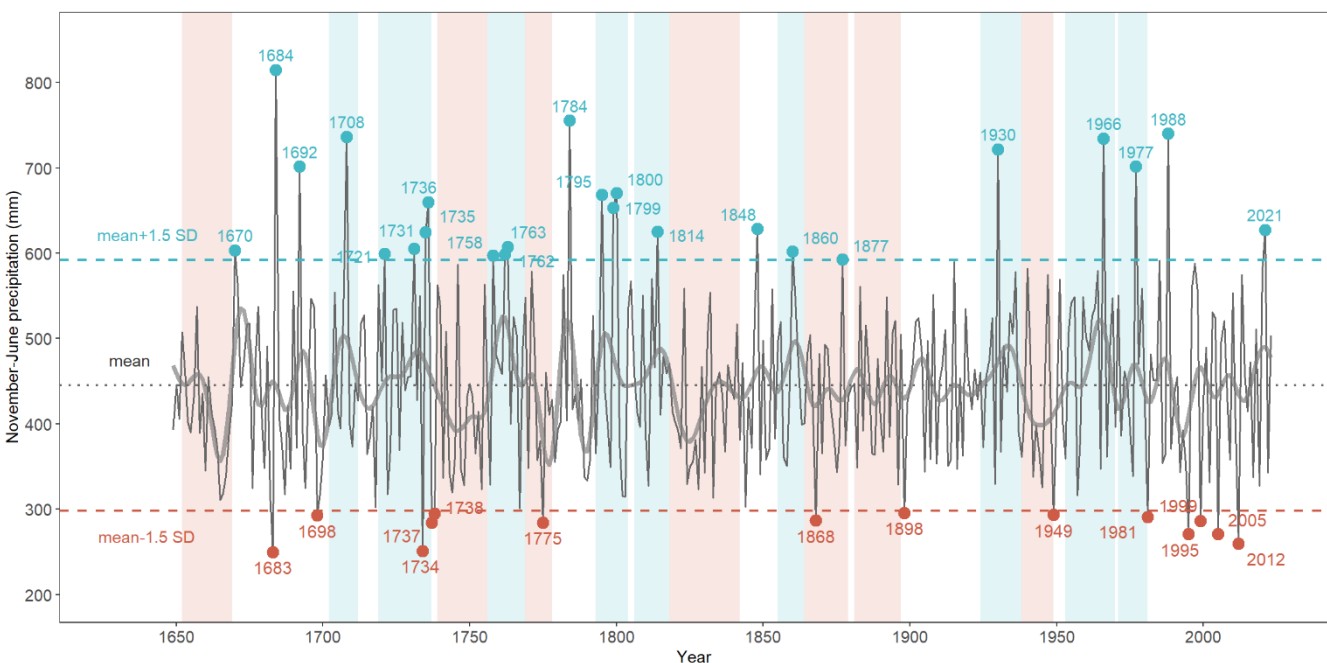


**Figure 6: Reconstructed November–June precipitation from 1649-2023 based on pollarded oak LWI data. Shaded areas indicate wet (blue) and dry periods (red), while extreme wet and dry years are marked with blue and red dots, respectively. Thick curve is a 10-year low-pass filter.**

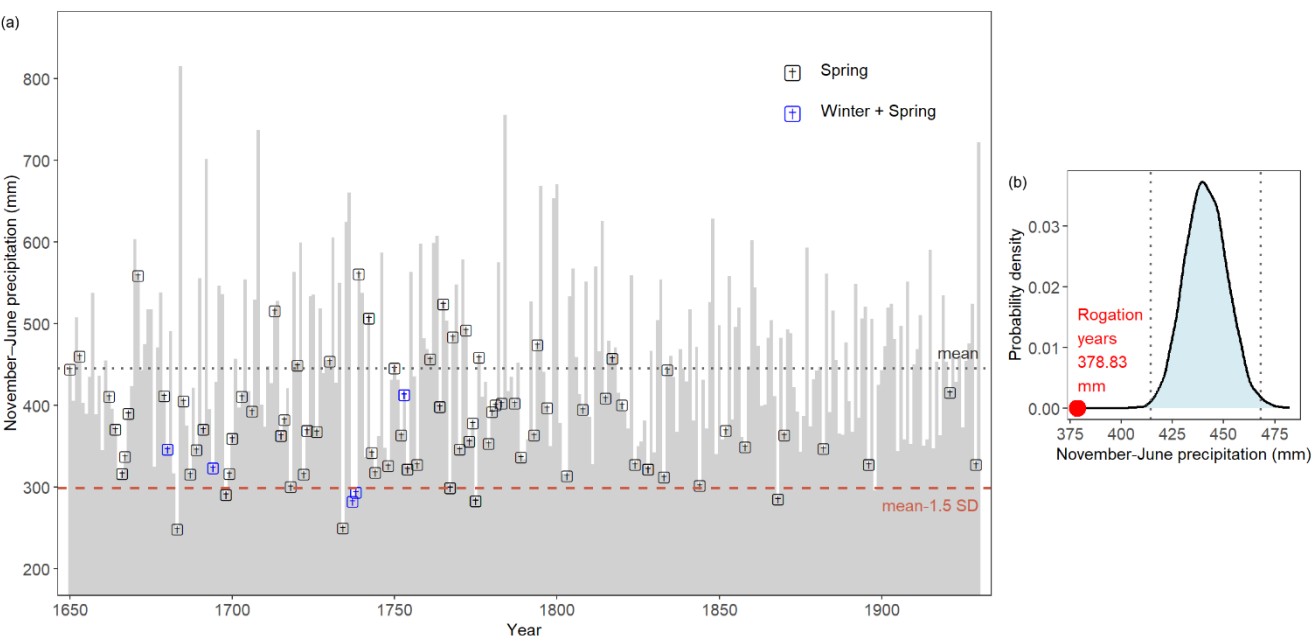






**Figure 7: (a) Comparison of reconstructed November–June precipitation (histogram) with *pro pluvia* rogation ceremonies (spring in black, winter & spring in blue). Multiple ceremonies in a year may overlap. (b) Mean estimated precipitation of the 81 rogation years from 1649–1929 (red dot) compared to the Monte Carlo simulation distribution, with dotted lines indicating 99% confidence intervals.**

## 4 Discussion

For the first time, we produced a 375-year precipitation reconstruction based on tree-ring data from managed trees in communal *dehesas* of northcentral Spain. The reconstructed precipitation series (November to June) accounts for more than 75% of the annual total, and up to 70% of this signal is retained in the latewood component, which emerged as the most robust proxy. The reliability of the reconstructed signal was further supported by independent historical documentary evidence, including rogation ceremonies. Despite the growth reductions after pollarding, we demonstrated that the stands retained a coherent climate signal, owing to the asynchronous management of individuals within each *dehesa*. This opens the possibility of using the extensive network of *dehesas* scattered across the Mediterranean lowlands to develop high-resolution climate reconstructions in regions traditionally underrepresented in dendroclimatic studies.

The chronology stands out for the strength of its hydroclimatic signal, with latewood width explaining up to 70% of the variability in seasonal precipitation. This level of explained variance equals or exceeds that reported in most existing oak-based reconstructions worldwide, including *Quercus douglasii* in California (51–70%, Gervais, 2006; Howard et al., 2023), *Q. infectoria* in Iran (50%, Azizi et al., 2013), and *Q. ilex* in southern Portugal (65%, Leal et al., 2015). In more humid temperate regions of Europe—such as Bohemia or southern England—explained variance declines markedly (≤34%, Dobrovolný et al., 2018; Wilson et al., 2013). This exceptional signal strength likely reflects both regional environmental constraints and reduced inter-tree competition. In dry-summer Mediterranean climates, soil water reserves are rarely fully replenished each year, making oak growth largely dependent on accumulated precipitation (Gallego et al., 1994; Hernández-Santana et al., 2008). Furthermore, the artificial arrangement of trees on relatively flat surfaces enhances their exposure to thermal extremes, potentially leading to persistently higher evapotranspiration demands compared to closed canopy or naturally structured forests.

Our reconstruction highlights hydroclimatic extremes in northcentral Spain, showing that the region has experienced abrupt shifts between exceptionally wet and dry conditions, as exemplified by the consecutive occurrence of the driest and wettest years in the series (1683: 250 mm; 1684: 815 mm of November–June precipitation). Moreover, the identification of prolonged dry (1818–1842, 25 years) and wet periods (1953–1981, 29 years) indicates that hydroclimatic variability also operated at multi-decadal scales, with potential implications for both ecological dynamics and historical human activities. Notably, reconstructed extremely dry years during the pre-instrumental period coincided with historical records, such as *pro pluvia* rogation ceremonies, which are recognized as robust proxies for past agricultural droughts (Domínguez-Castro et al., 2008). In central Iberia, cereal crops and livestock constituted the main staples for population subsistence, and their productivity depended directly on soil moisture, which is largely determined by precipitation levels (Vicente-Serrano et al. 2020). During drought years, *pro pluvia* rogations served as institutional mechanisms to address social stress triggered by the threat of harvest




failures—events that often had far-reaching consequences, including malnutrition, market disruptions, and public health crises (Barriendos, 2005). Interestingly, some spring ceremonies coincided with years that were overall wet years in our reconstruction. These departures may reflect abundant rainfall that occurred after the rogation was held (Domínguez-Castro et al. 2012). However, their distribution was not random, as they were concentrated in Zamora during spring in the 17th century. The agreement between reconstructed drought extremes and documentary evidence underscores the robustness of our

chronology, a conclusion further strengthened by its consistency with previous large-scale reconstructions.

Our reconstruction showed the strongest agreement with the Great Eurasian Drought Atlas (Cook et al., 2024), following by regionally focused hydroclimate reconstructions, particularly the Western Mediterranean precipitation index (Camuffo et al., 2010), the Andalusia DJF precipitation Index (Rodrigo et al., 1999), and the Ebro Valley drought Index (Tejedor et al., 2019). The high correlation with the latter is somewhat unexpected, given that the Ebro Basin lies outside the core spatial domain of

our chronology (Fig. 2). However, since the Ebro Valley Drought index is based on documentary records of historical droughts, it likely captures broader hydroclimatic anomalies extending beyond its immediate geographical boundaries, which may explain its strong coherence with our precipitation series. In contrast, our reconstruction showed weaker correlations with large-scale products, such as the European gridded series by Pauling et al. (2006) and the global PHYDA reconstruction (Steiger et al., 2018). This is due to their lack of regional/seasonal skill, which limits their capacity to resolve localized

hydroclimatic variability. As such, pollarded oaks from northcentral Spain offer additional paleoclimatic information that captures interannual to decadal precipitation variability with high regional fidelity, serving as a valuable benchmark to refine and strengthen hydroclimate assessments beyond what is possible with previously available large-scale products.

In this regard, latewood was the most suitable proxy for hydroclimate reconstruction, as it is the tree-ring component most sensitive to precipitation while still preserving a coherent climatic signal at the chronology level despite recurrent pollarding.

Under stress conditions, earlywood is prioritized due to its hydraulic function, whereas latewood—primarily involved in mechanical support—is more expendable (Domec and Gartner, 2002). Consequently, earlywood production remained relatively sTable Sfter pollarding, buffering variability in total ring-width (Bernard et al., 2006; Rozas, 2005; Sanmiguel-Vallelado et al., 2024), while latewood formation was significantly reduced. This led to a temporary decline in the climatic signal captured by latewood at the tree level during the first three years post-pollarding. However, from the eighth year onward,

the latewood index, together with the total ring-width index, exhibited enhanced sensitivity to precipitation compared to the pre-pollarding period. This pattern suggests that, as trees recover from pollarding, their growth becomes more responsive to water availability. Such a response aligns with previous studies that report growth stimulation following pollarding, largely due to enhanced light availability (Cañellas et al., 2004; Cutter et al., 1991; Mayor and Rodà, 1993; Tonelli et al., 2023). However, this greater sensitivity may also imply an increased vulnerability to intense or prolonged droughts. Importantly, our

results indicate that the impact of pollarding at the individual level did not compromise the integrity of the stand-level chronology. This resilience likely stems from two complementary factors: the post-pollarding enhancement of the climatic signal and the historical asynchrony in pollarding among trees, where cuts occur at different times among trees and owners

(Olano et al. 2023). In this context, unpollarded individuals may have compensated for the temporary loss of climatic signal in trees undergoing management, particularly given the low frequency of pollarding events (Sanmiguel-Vallelado et al. 2024).

The climatic sensitivity of Mediterranean deciduous oaks is well documented (Alla and Camarero, 2012; Camarero et al., 2024; Leal et al., 2015; Romagnoli et al., 2018; Sánchez-Salguero et al., 2020; Tessier et al., 1994). However, pollarded individuals within *dehesa* systems have received comparatively little scientific attention, despite recent studies highlighting the exceptional strength of their climate signal (Camarero and Valeriano, 2023; Olano et al., 2023). In this study, we unlock their full dendroclimatic potential for the first time, demonstrating that these trees retain a strong and coherent hydroclimate

signal over centuries, even under the repeated disturbances associated with traditional management. Communal *dehesa*s are relatively widespread across the Mediterranean Basin and offer a valuable opportunity to reconstruct long-term precipitation variability across different regions. Such reconstructions could significantly improve our understanding of current climate change dynamics, particularly for precipitation—one of the most uncertain components in future climate scenarios (Deser et al., 2012). Recognizing the dendroclimatic value of communal *dehesas* adds relevance to these landscapes of exceptional

ecological and cultural significance (Olano et al., 2025). Preserving and studying these systems is therefore both essential and urgent, as the abandonment of traditional management, combined with increasing drought stress—especially affecting large, old trees (Pennisi, 2019)—has already led to widespread dieback (Colangelo et al., 2024).

**5 Conclusions**

Our findings demonstrate that deciduous pollarded oaks in communal *dehesas* constitute a reliable dendroclimatic archive,
capable of capturing interannual to decadal precipitation variability over centuries in regions traditionally underrepresented in dendroclimatic research. Latewood is the most suitable proxy, retaining a strong and stationary hydroclimate signal at the chronology level despite temporary growth reductions following pollarding. When sampled from stands where cuts occur at different times among trees, these short-term, post-pollarding growth effects do not synchronize across individuals, and accounting for pollarding signatures has negligible impact preserved climate signals. Reconstructions combining such data at

the landscape level provide unique information on *(i)* ecological and cultural heritage, and *(ii)* past hydroclimate variability in lowland mediterranean environments highly vulnerable to climate change.

**Code availability**

Code will be made available upon request.

**Data availability**

Data will be made available upon request.

**Author contribution**

JMO, GS and AS conceptualized the study. Fieldwork was conducted by JMO, GS, and AS. Cross-dating was performed by JMO. Data curation was carried out by AS and JMO. Instrumental climate data (FIC) was provided by CP. AS performed the formal analysis under the guidance of MCAT. AS prepared the manuscript with contributions from all co-authors. Visualizations were created by AS and MCAT. Supervision was provided by JMO, GS, MCAT, and JE. Funding was acquired by JMO and GS. The project was administered by JMO and GS. All authors have read and agreed to the published version of the manuscript.

**Competing interests**

The authors declare that they have no conflict of interest.

**Acknowledgements**

We are grateful to the "Servicio de Medio Ambiente de la Junta de Castilla y León," and in particular to José Antonio Lucas, for granting the sampling permits. We also thank the local managers and environmental agents for sharing valuable information regarding the study sites and their management practices. We deeply appreciate the support of Juan Carlos Rubio and Alfonso Martínez, whose assistance was essential for sample preparation. Fieldwork and lab work were made possible thanks to the generous help of the members of Cambium research group. We also wish to express our sincere thanks to all members of Jan Esper's research group at Johannes Gutenberg Universität Mainz, whose insightful comments greatly contributed to the development of this publication.

**Financial support**

This research was funded by the project GIANTS (PID2023-147214NB-I00) funded by MICIU/AEI/10.13039/501100011033 and FEDER, UE. Alba Sanmiguel-Vallelado was supported by the postdoctoral grant JDC2022-048316-I, funded by MICIU/AEI/10.13039/501100011033 and by the European Union NextGenerationEU/PRTR. She also received funding from the "Movilidad Investigadores e Investigadoras UVa-Banco Santander 2024" program, which supported a research stay at the of Jan Esper's research group at Johannes Gutenberg Universität Mainz, Germany.

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
