# Peer review of "Unlocking the potential of pollarded oaks: A 375-year hydroclimate reconstruction from northcentral Spain"

_EGUsphere, 2025_

## Author Comment (AC1)

- Line 54: "the upper part of the tree" I think that it would be better to say that part of the canopy is removed.

  We agree with the reviewer, the revised text now reads: "Pollarding involves removing part of the tree canopy through systematic pruning at varying intensities (Petit and Watkins, 2003)."

- Figure 1. You have Quercus pyrenaica over the sea... is there any better map for this species?

  Thanks for the remark. We have adjusted the *Quercus pyrenaica* distribution to the terrestrial area. Accordingly, the figure caption now specifies that the map is "adapted from Caudullo et al. (2017)."

- Detrending: "smoothing splines with a 50% frequency cutoff set to two-thirds of the series length" I believe that this is a bit tricky detrending option since the stiffness of your spline changes from sample to sample and therefore one retain different climatic information (see Klesse 2021). I would suggest to redo the analyses while keeping a fixed frequency cutoff for the splines.

  We agree with the reviewer that standardization of tree-ring data needs to be approached with caution, and we share many of the sentiments of the Klesse paper (although we believe the issue is most problematic when dealing with multidecadal variability or trend). However, we also believe that any of such decisions made are subjective and that arguments can be put forth for many different detrending choices. Nonetheless, the raw latewood data was detrended again, using a fixed frequency cutoff, and the relationship with the chronology presented in the first submission, as well as with the target precipitation data, is presented below (**Table R1**).

**Table R1**. Correlation between fixed-frequency and ADCS chronologies (power transformed and non-power transformed) and original iteration (for 1902-2020 / full chronology), and the different climate products.

| | Presented chronology | FIC, 1952-2020 | CRU, 1952-2020 | CRU, 1902-2020 |
|---|---|---|---|---|
| **Fixed-50, PT** | 0.958 / 0.927 | 0.820 | 0.794 | 0.746 |
| **Fixed-50, non-PT** | 0.957 / 0.925 | 0.810 | 0.790 | 0.748 |
| **ADC, PT** | 0.928 / 0.834 | 0.803 | 0.799 | 0.733 |
| **ADC, non-PT** | 0.930 / 0.829 | 0.815 | 0.777 | 0.747 |

Although minor differences are evident, these are negligible when considering other sources of uncertainty. As such, we feel comfortable with keeping the original chronology as the main predictor of our reconstruction.

- Figure 2. I do not think that the points on the maps regarding the pro pluvia events are necessary here. I would add them in figure 1.

  The points representing *pro pluvia* events have been removed from Figure 2 and are now shown in Figure 1a. The corresponding references to these figures in the text have been updated accordingly.

- Lines 89: there are some typos there.

  We carefully reviewed line 89 as well as all the entire Figure 1 legend, and we made adjustments to improve clarity and style. The revised sentence now reads: "(b) Location of the Vilviestre (Vi) and Valonsadero (Va) *dehesas*, northwest of Soria, as well as the central position of the grid cells from the Climate Research Unit (CRU) and *Fundación para la Investigación del Clima* (FIC) used to obtain climate data. Digital Terrain Model source: PNOA 1:200,000, ETRS89 HU30, Soria."

  Since the comment number may have been affected by an error in the line indication, we also checked lines 189 and 289. No typos were found in line 189. In line 289 (Table 2), we detected minor clarity issues and edited the caption accordingly. The updated caption reads:

  "Table 2: Statistical correlations between the new November–June precipitation reconstruction and previous reconstructions of hydroclimate records. Correlations are calculated for the whole period (since 1649) and separately for each century. Pearson correlation is used in all cases, except for Tejedor et al. (2019), where a non-parametric Spearman correlation was applied. Significant values ($p < 0.05$) are indicated with an asterisk (*). IO refers to instrumental observations, DS to documentary sources, NP to natural proxies, and PS to the PHYDA paleoclimate series (reanalysis from multiple proxies and climate-model assimilation; Steiger et al., 2018). SPEI stands for the Standardized Precipitation Evapotranspiration Index, DJF for December–February accumulated precipitation, MAM for March–May, and JJA for June–August. NA indicates missing century-scale correlations due to a shorter common period between series."

- Line 387: historical asynchrony: this is what is missing a bit here. Maybe the authors have already published about it somewhere else but since it is one of the main reasons that your reconstruction is kind of robust, this needs to be also be shown here.

  We agree that the discussion benefits from explicitly illustrating historical asynchrony. We have added a new supplementary figure (Fig. S4), referenced in line 201, which illustrates the number of trees affected by pollarding through time, following the methodology of Sanmiguel-Vallelado

et al., 2024 (M&M; L. 139-141). This figure shows that pollarding is markedly asynchronous across trees in both *dehesas*, with most events affecting only a small fraction of the available trees (75% of events involve <6% of trees). Only three events exceed 30% synchrony: 1697 (36.4%), 1736 (30.6%), and 1800 (31.7%). In all three cases, the tree ring following pollarding is narrow, which is unlikely to be an artifact, as the years 1698 and 1737 are supported by documented rogative events.

[Figure]

**Figure S4. Percentage of trees affected by pollarding events each year (grey segments) relative to the number of trees available in the chronology at that time (green line), based on Sanmiguel-Vallelado et al. (2024).**